# Giant single-crystal-to-single-crystal transformations associated with chiral interconversion induced by elimination of chelating ligands

Yun Li[1], Bo Zhao[1], Jin-Peng Xue [1], Jing Xie [1], Zi-Shuo Yao [1✉] & Jun Tao [1✉]

Numerous single crystals that exhibit single-crystal-to-single-crystal (SCSC) transformations have been reported, and some of them show great promise for application to advanced adsorption materials, magnetic switches, and smart actuators. However, the development of single crystals with super-adaptive crystal lattices capable of huge and reversible structural change remains a great challenge. In this study, we report a $Zn^{II}$ complex that undergoes giant SCSC transformation induced by a two-step thermal elimination of ethylene glycol chelating ligands. Although the structural change is exceptionally large (50% volume shrinkage and 36% weight loss), the single-crystal nature of the complex persists because of the multiple strong hydrogen bonds between the constituent molecules. This allows the reversible zero-dimensional to one-dimension and further to three-dimensional structural changes to be fully characterized by single-crystal X-ray diffraction analyses. The elimination of chelating ligands induces a chiral interconversion in the molecules that manifests as a centric-chiral-polar symmetric variation of the single crystal. The study not only presents a unique material, featuring both a periodic crystal lattice and gel-like super-ductility, but also reveals a possible solid-state reaction method for preparing chiral compounds via the elimination of chelating ligands.

---

[1] Key Laboratory of Cluster Science of Ministry of Education, School of Chemistry and Chemical Engineering, Beijing Institute of Technology, Beijing 100081, People's Republic of China. ✉email: zishuoyao@bit.edu.cn; taojun@bit.edu.cn

Single crystals that possess adaptive crystal lattices in response to external stimuli are garnering increasing research attention[1–8]. Such structural flexibility provides a practical way to access superior adsorption materials, materials with tunable magnetic properties, ferroelectric materials, and efficient actuators[9–19]. Typical examples are single crystals that undergo single-crystal-to-single-crystal (SCSC) transformations induced by concerted guest sorption or reactions between the guest and host structures, which are reminiscent of those in microtubules and enzymes in biological assemblies[20–30]. The direct structural information obtained from single-crystal X-ray diffraction (SC-XRD) analyses may enhance our understanding of the molecular transport and reactions in these condensed phase systems. However, the SCSC transformations in typical single-crystal materials are primarily small because of the rigid chemical bonds and spatial constraint in the crystal lattice. Recently, high-grade structural transformations involving large shifts in mass and volume have been observed in several porous single crystals as a consequence of guest sorption[31–33]. Such counterintuitive phenomena are usually associated with special interactions between the material's constituent molecules.

In addition to the guest sorption by porous structures, the elimination of chelating ligands from a nonporous crystal might also lead to a dramatic SCSC transformation because two coordination sites are released concertedly during the reaction. Moreover, the synchronous shift of *cis*-coordination sites facilitates the conformational conversion of the molecules, potentially affording single crystals with intriguing chiral and optical properties[34]. However, to the best of our knowledge, SCSC transformation stimulated by the elimination of chelating ligands has not yet been demonstrated because of the strong coordination ability of chelating ligands and the fact that such large structural variation disrupts the periodic structure of the crystal.

Herein, we report that a simple single crystal of $[Zn^{II}(eg)_3]SO_4$ (eg = ethylene glycol) can undergo two-step reversible solid-state structural variation by successive elimination of eg ligands upon heating. The elimination of the chelating ligands leads to giant SCSC transformations with a total shrinkage of 50% in volume and 36% in mass. Serendipitously, although the structural change is exceptionally large, the single-crystal nature persists because of the strong hydrogen-bond interactions between the complex cations and sulfates. Therefore, the remarkable SCSC transformations in which the zero-dimensional structure $[Zn^{II}(eg)_3]SO_4$ (**1-0D**) shifts to the one-dimensionally structured $[Zn^{II}(eg)_2(\mu\text{-}SO_4)]_n$ (**1-1D**), and further to the three-dimensionally structured $\{[Zn^{II}(eg)_2]Zn^{II}(\mu_3\text{-}SO_4)_2\}_n$ (**1-3D**) are unambiguously characterized by SC-XRD structural analyses. Moreover, the SCSC transformation involves a chiral interconversion in the molecular complex, which manifests as centric-chiral-polar symmetry variations in the single crystals.

## Results

**Structural characterization of 1-0D.** The colorless single crystal of **1-0D** was prepared by a slow diffusion of acetone solvent into an eg solution of $ZnSO_4·7H_2O$. SC-XRD analyses at 293 K revealed that the $Zn^{II}$ metal center is chelated by three eg ligands in a twisted $\Lambda$ or $\Delta$ conformation (Fig. 1 and Supplementary Fig. 1). The $SO_4^{2-}$ dianions that act as the counteranions are alternatively connected with complex cations through electrostatic interaction and strong O−H···O hydrogen-bond interactions (Supplementary Fig. 2)[35]. Because the complex cations in the $\Lambda$ and $\Delta$ conformations are arranged in an AABB manner along the crystallographic c axis, the compound crystallizes in the centrosymmetric monoclinic space group $P2_1/c$ with lattice parameters of $a = 8.7864(4)$ Å, $b = 7.6102(3)$ Å, $c = 19.3284(10)$ Å, and $\beta = 99.475(5)°$ (Z = 4).

Thermogravimetric (TG) measurement showed that **1-0D** undergoes a two-step weight loss upon heating the crystals to 430 K (Fig. 2a). The weight loss of ~18% in the first step (344–378 K) and further ~18% (390–410 K) in the second step (for a total of 36%) indicate that two of the three eg ligands are removed one-by-one in the two-step weight loss process (the corresponding calculated weight-loss values are 18% in each step). The elimination of eg ligands is highly dependent on the heating process. As shown in Supplementary Fig. 3, when the heating rate is faster than 10 K/min, the first-step weight loss is largely suppressed. The material maintains its single-crystal nature upon the large two-step weight loss (Supplementary Fig. 4), presenting the opportunity to investigate these huge structural transformations at the atomic level using SC-XRD analyses.

**Structural characterization of 1-1D.** Upon heating the crystal at 366 K for 2 h, the single crystal adopts a one-dimensional (1D) structure (Supplementary Fig. 5). The in situ diffraction analyses revealed that one chelating ligand of each $[Zn^{II}(eg)_3]^{2+}$ complex cation is removed and the corresponding *cis*-coordination sites are substituted by two O atoms from two neighboring sulfate molecules in the first-step SCSC transformation (Fig. 1b and Supplementary Fig. 6). Therefore, the discrete metal centers in **1-0D** are bridged by the bidentate sulfate dianions into the 1D chain structure of **1-1D** with the helices running along the crystallographic a axis[36]. Because the helices in each individual single crystal have the same handedness (Supplementary Fig. 7), the single crystal adopts the orthorhombic Sohncke space group $P2_12_12_1$[37]. Furthermore, the elimination of eg ligands induces a chiral inversion in the coordination sphere of the metal centers. As shown in Fig. 1b and Supplementary Fig. 8, the twisted $\Lambda$ and $\Delta$ conformations of the complex cations in **1-0D** adopt an identical *cis*-$\Lambda$ configuration (or *cis*-$\Delta$ in an enantiomeric crystal) in **1-1D**. The SCSC transformation introduces huge variations in the lattice parameters, with $a = 6.4536(5)$ Å, $b = 9.8866(6)$ Å, and $c = 15.1208(9)$ Å (Z = 4) for **1-1D**, corresponding to a large volumetric shrinkage in the order of 25%.

**Structural characterization of 1-3D.** The eg ligands can be further eliminated upon heating to ~410 K, leading to a second-step SCSC transformation. As shown in Fig. 1c and Supplementary Fig. 9, half of the eg ligands in **1-1D** are eliminated in the second-step SCSC transformation. Correspondingly, the 1D structure is further converted into a three-dimensional (3D) network. As shown in middle panel of Fig. 1c, the asymmetric unit of the new structure comprises four $Zn^{II}$ metal centers. Zn1 and Zn4 retain $O_6$ coordination spheres with four O atoms from two eg ligands and two O atoms from two sulfate anions, while Zn2 and Zn3 adopt a tetrahedral coordination sphere with four O atoms from four sulfate anions. The $Zn^{II}$ ions in hexa- and tetra-coordination spheres bridged by two sulfate anions form dinuclear circular building blocks that are further linked into a 3D network via Zn−O coordination bonds. In **1-3D**, every sulfate dianion behaves as a tridentate ligand that connects with one hexacoordinated $Zn^{II}$ ion and two tetracoordinated $Zn^{II}$ ions. Because all the eg ligands in the 3D structure orient principally in a same direction along the crystallographic b axis, the symmetry of the single crystal changes to that of the polar Sohncke space group $P2_1$, with a spontaneous polarization emerging along the crystallographic b axis. During the second-step SCSC transformation, the lattice parameters undergo a further huge variation with $a = 10.0682(5)$ Å, $b = 9.9679(4)$ Å, $c = 12.7621(6)$ Å, and $\beta = 90.093(4)°$ (Z = 8), representing another 25% volumetric shrinkage (on the basis of the initial volume of **1-0D**). Remarkably, the single crystal of **1-0D**

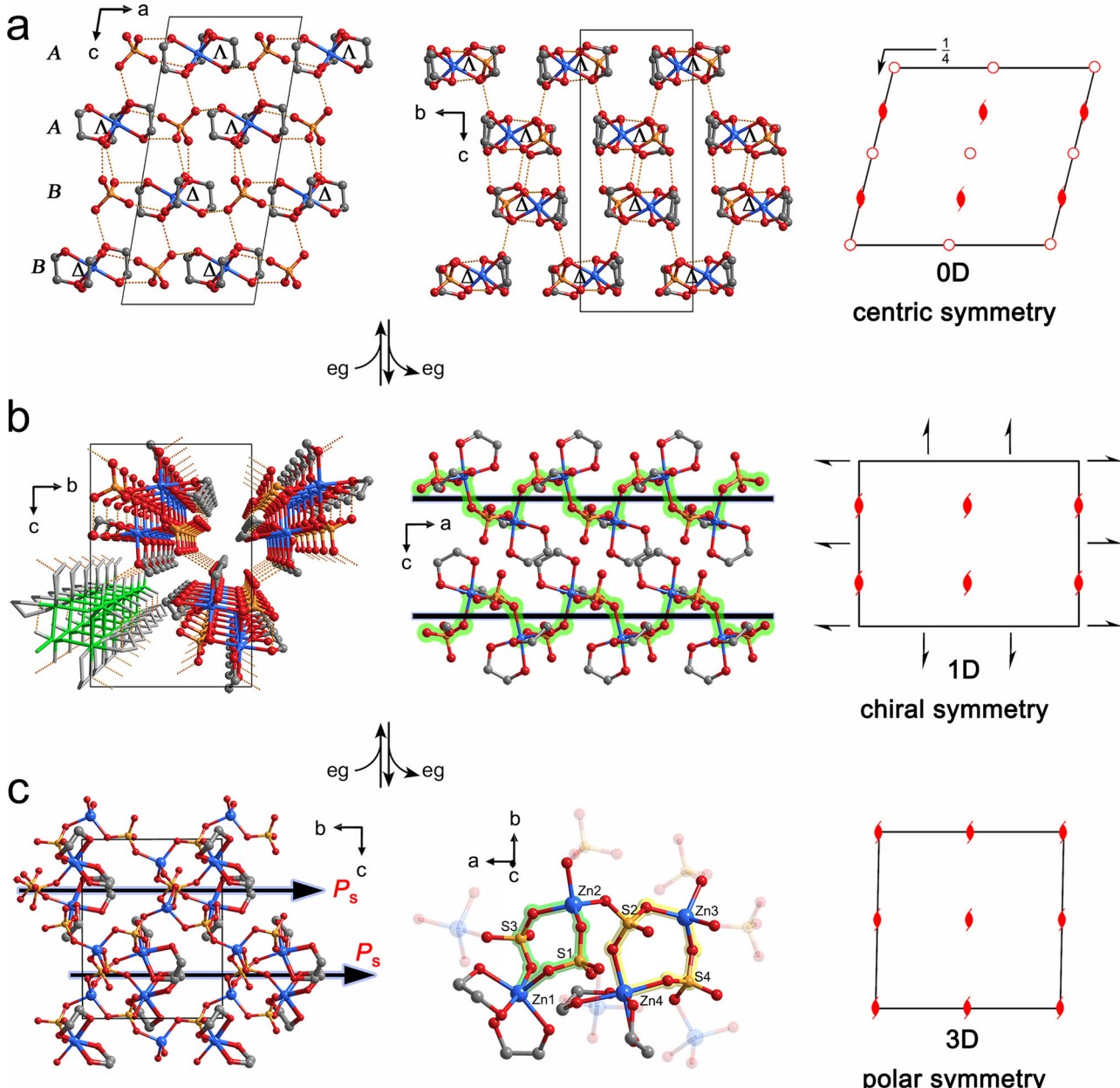

**Fig. 1 Two-step reversible SCSC transformation associated with symmetry variation. a** The single-crystal structure of **1-0D**. In the crystal, the Zn^II ions coordinated by three eg ligands in Λ or Δ configurations manifest as discrete metal complexes that are connected by sulfate dianions through strong hydrogen bonds. The complex cations in Λ and Δ configurations are arranged in AABB mode along the crystallographic *c* axis. **b** The single-crystal structure of **1-1D**. One of the three eg ligands is replaced by two sulfate dianions during the first-step SCSC transformation. Correspondingly, the Zn^II ions are linked into 1D helical structures that extend along the crystallographic *a* axis. **c** The single-crystal structure of **1-3D**. After further elimination of one-third of the eg ligands, the structure transforms into a 3D network with a spontaneous polarization emerging in the crystallographic *b* axis. The dashed lines denote the O–H···O hydrogen-bond interactions. Zn, blue; O, red; C, gray; S, yellow.

can be recovered by immersing **1-3D** in an acetone solution of eg (Supplementary Table 1).

The SCSC transformation was further investigated by powder X-ray diffraction (PXRD) analyses. As shown in Fig. 2b, the peaks representing the structures of **1-1D** and **1-3D** emerge as new peaks upon heating the material to 368 and 398 K, respectively. The sharp diffraction peaks indicate that the high-quality crystal lattice is maintained after the two-step transformation. The transition temperature in the PXRD measurement is slightly lower than that in the TG analyses, which can be ascribed to the different heating method and test condition in the PXRD and TG measurements (for details see the "Methods" section). Moreover,

the reversibility of the structural transformations was also monitored by in situ PXRD by immersing the polycrystals of **1-3D** into an acetone solvent containing eg ligands. The PXRD analyses present peaks that are assigned to the structures of **1-1D** and **1-0D** after 180 and 720 min, respectively, further confirming the reversible nature of the two-step SCSC transformation (Supplementary Fig. 10).

The variation of crystallographic symmetry during the SCSC transformation was verified through a temperature-dependent second harmonic generation (SHG) measurement, which is highly sensitive to the centrosymmetric-to-noncentrosymmetric structural change[38]. As shown in Fig. 3a, the SHG signal appears abruptly upon

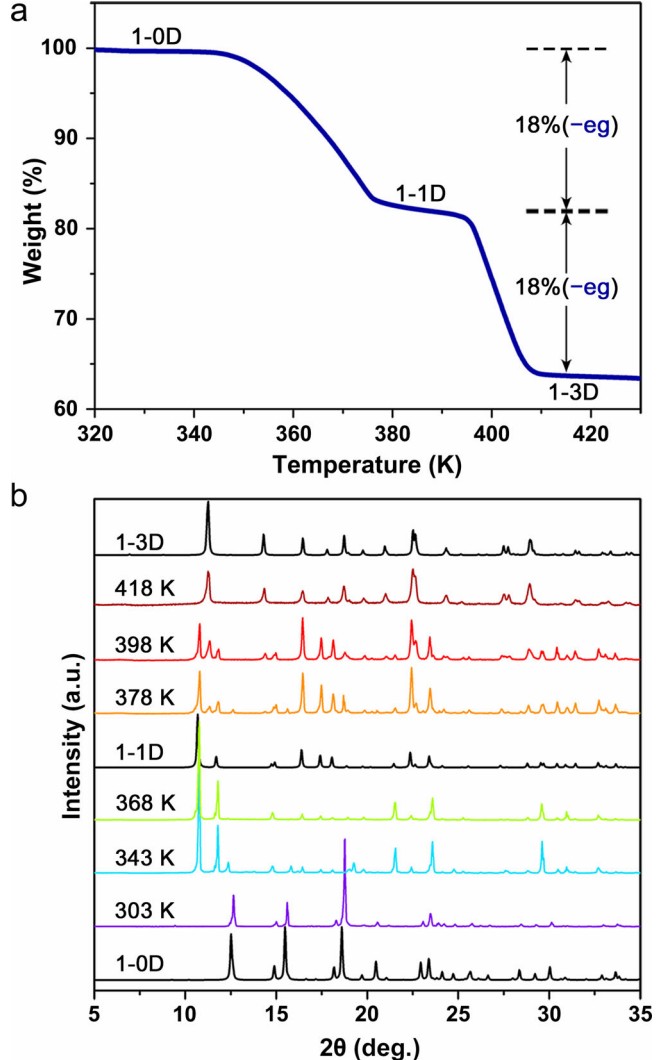

**Fig. 2 Crystal transformation monitored by TG and PXRD measurements.**
**a** TG curve of compound **1-0D** with a heating rate of 3 K min⁻¹. The weight losses of 18% at 380 K and 36% at 410 K suggest that two-thirds of the eg ligands are eliminated in the two-step transformation upon heating. **b** Variable-temperature PXRD of **1-0D**. The peaks corresponding to **1-1D** and **1-3D** appear at 343 and 378 K, respectively.

heating the polycrystals to ~378 K, confirming the centric-to-acentric structural change of the crystal during the SCSC transformation. Moreover, the chirality of individual single crystals of compound **1-1D** was further examined by solid-state circular dichroism (CD) spectroscopy. As shown in Fig. 3b and Supplementary Fig. 11, the crystal of **1-1D** with Λ molecular conformation exhibits positive Cotton effects at *ca.* 237 and 435 nm and a negative effect at 311 nm, confirming the chirality of **1-1D**, and the near-mirror images observed for the crystal of **1-1D** with Δ molecular conformation verify the enantiomeric nature of **1-1D**.

## Discussion

Reports of reversible SCSC transformations associated with large volume and mass changes are rare in the literature (Supplementary Table 2). The 50% total volumetric shrinkage and 36% total weight loss observed for the present material, which are reminiscent of the expansion of clay structures by adding ethylene glycol[39,40], are larger than those observed for other typical single-crystal materials that undergo reversible SCSC transformations, and they are even comparable to those of two recently discovered

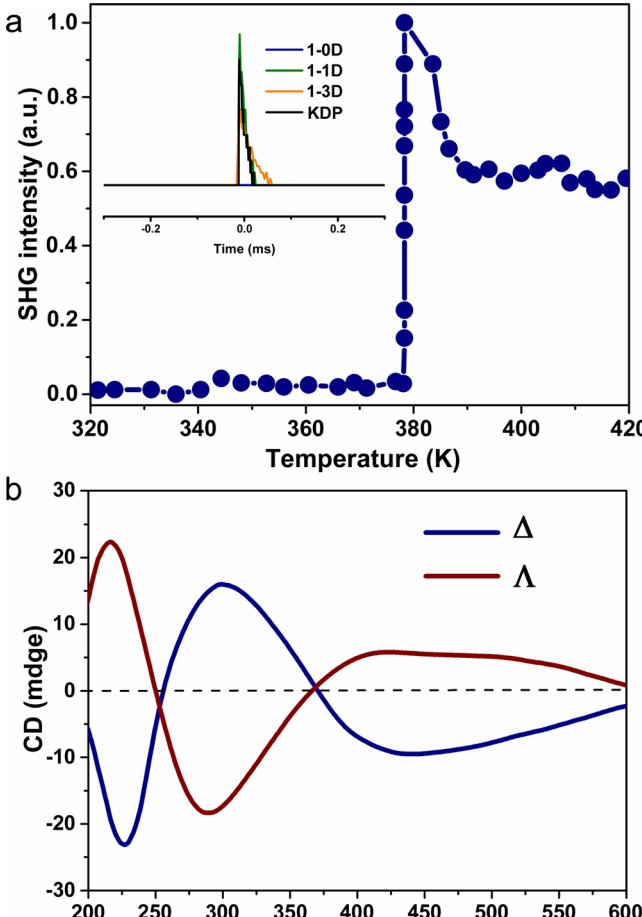

**Fig. 3 The centric-to-acentric symmetric variation involved in the two-step structural transformation. a** The temperature-dependence of SHG upon heating the crystal of **1-0D** to 420 K. Inset: comparison of the SHG signal with KDP. **b** The CD spectra of **1-1D** with enantiomeric structures.

porous crystalline materials that contain biomolecules, i.e., ferritin and tripeptide, that demonstrate giant expandability in response to guest sorption[31,32]. Although detailed atomic level structural information is important for understanding these super-adaptive crystal lattices, such information is difficult to obtain for these biomolecules because of the disordered nature of the guests in their pores and the disruption of the single-crystal lattice that occurs upon giant expansion. However, the simple but robust crystal lattice of the current material presents a unique opportunity to rationally investigate and further understand its giant structural transformation and accompanying symmetric variation.

Our detailed structural investigations revealed the presence of strong hydrogen-bond interactions in the structures both before and after the SCSC transformation. As shown in Supplementary Figs. 2 and 6 and Supplementary Tables 3 and 4, the O···O distances are in the range 2.610–2.676 Å in **1-0D** and 2.644–2.877 Å in **1-1D**. In the crystal of **1-0D**, which crystallizes in the space group $P2_1/c$, the complex cations and sulfate dianions are hydrogen-bonded into 1D helical architectures that screw along the crystallographic $C_2$ axes (in the direction of the $b$ axis, see Figs. 1a and 4). Because the 1D helices run along the crystallographic $a$ axis in **1-1D**, which corresponds to the crystallographic $b$ axis in **1-0D** according to the results of crystal face indexing (Supplementary Fig. 12), we propose that the sulfates in

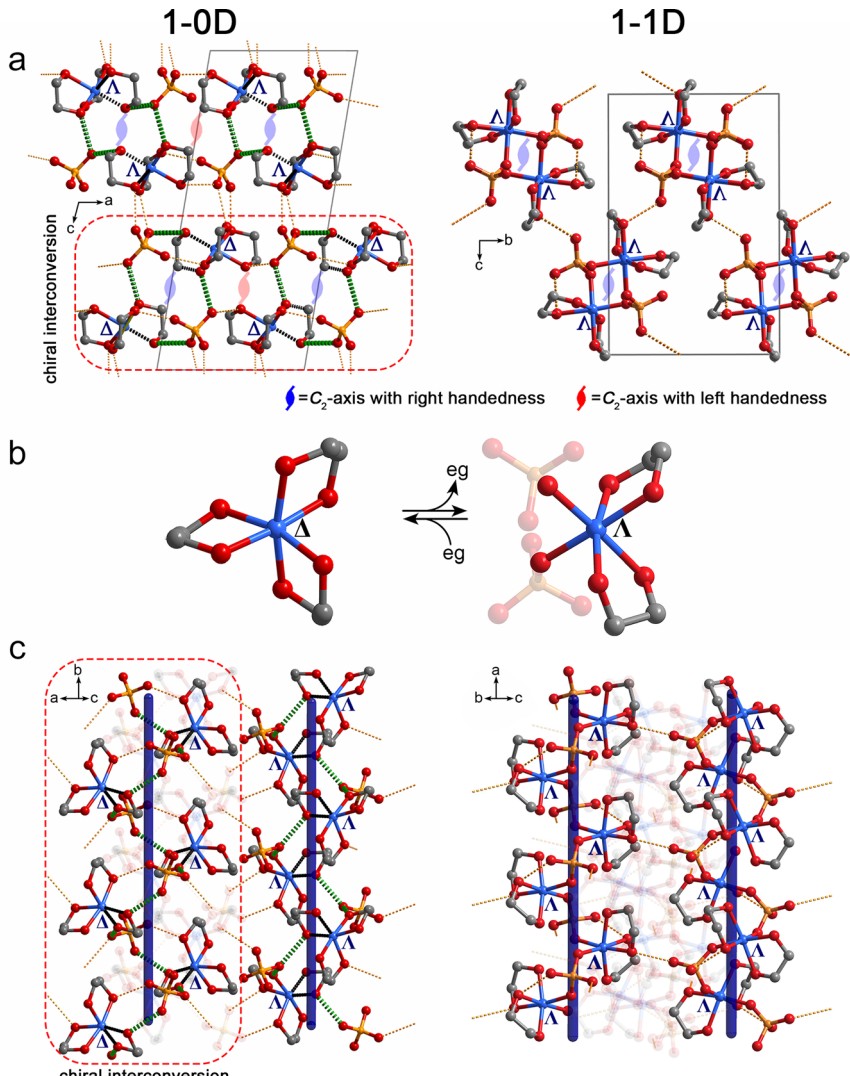

**Fig. 4 Proposed mechanism for the SCSC transformation from 1-0D to 1-1D. a** Contraction of the structure in the *ac*-plane. In the crystal of **1-0D**, the hydrogen-bonded complex cations and anions form 1D helical architectures possessing both left and right handedness around the crystallographic $C_2$ axes (left panel), while the 1D helices in **1-1D** exhibit identical handedness (right panel). **b** Chiral interconversion of complex cation with Δ configuration during the transformation from **1-0D** to **1-1D** with right handedness. **c** Contraction of structure along the crystallographic *b* axis. The green and yellow dashed lines denote the strong O–H···O hydrogen-bond interactions. The sulfates should substitute the eg ligands along the bold green dashed lines. Zn, blue; O, red; C, gray; S, yellow.

**1-0D** approach the metal centers and replace the eg ligands along the hydrogen bonds in the direction of the *b* axis and then link the discrete complex cations into a 1D helix. Accordingly, the crystallographic $C_2$ axes in **1-0D** are adopted in the new structure of **1-1D** (Fig. 4a). However, the centrosymmetric structure of **1-0D** suggests the 1D hydrogen-bonded architectures manifest both left- and right-handedness, while the 1D helices in **1-1D** possess identical handedness. Therefore, the condensation reactions between the $[Zn^{II}(eg)_3]^{2+}$ complex cations and sulfates should occur around the $C_2$ axes with the same handedness. As a consequence of the competition between the left- and right-handedness, single crystals of **1-1D** with enantiomeric handedness are produced during the SCSC transformation (Supplementary Fig. 13)[41,42]. The strong hydrogen-bond interactions not only spatially define the motion of the sulfate dianions, they also play pivotal roles in fastening the crystal lattice during the giant structural transformation[22,31,32]. Therefore, the single-crystal structure is maintained after elimination of the chelating ligands. Furthermore, the relative motion of the sulfate dianions

realizes a chiral interconversion in half of the complex cations through strong hydrogen-bond interactions. As shown in Fig. 4b, the Δ conformation of the complex cations in **1-0D** changes to the Λ conformation in the crystal of **1-1D** with right-handed helices. We believe that the large free volume in *cis*-coordination sites facilitates the conformational change of the molecular complex in response to the motion of sulfate anions. The chiral interconversion of the complex cations disrupts the centrosymmetry of the crystal. Consequently, the space group changes from $P2_1/c$ in **1-0D** to $P2_12_12_1$ in **1-1D**.

In the second-step SCSC transformation that occurs upon further heating, the 1D helical structure transforms to a 3D network with the 1D backbone retained along the crystallographic *a* axis in **1-3D** (Fig. 1 and Supplementary Fig. 14). Based on the structural motifs identified, the circular building block in **1-3D**, i.e., one hexacoordinated $Zn^{II}$ ion and one tetracoordinated $Zn^{II}$ bridged by two sulfate anions, can be produced within the helix as a result of the O atom from the sulfate dianion in the adjacent thread shifts toward the $Zn^{II}$ center and substitutes the eg ligand.

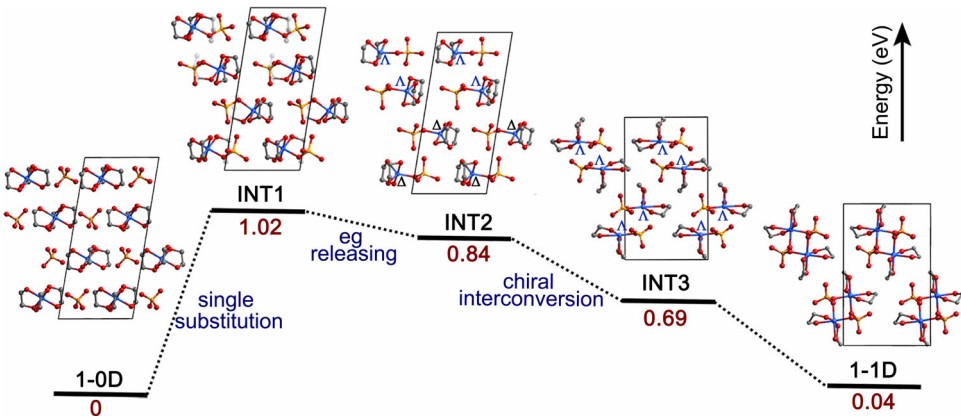

**Fig. 5 Density functional theory (DFT) calculated energy diagram for conversion from 1-0D to 1-1D.** INT1: one of the Zn−O(eg) bond was substituted by a sulfate ion. INT2: one eg ligand was released and the Zn(II) was coordinated by five O atoms. INT3: the Zn-motif underwent a chiral interconversion. INT denotes the intermediate state. Zn, blue; O, red; C, gray; S, yellow.

The dinuclear block provides two O coordination atoms and two $Zn^{II}$ coordination sites to further connect the other four building blocks. Consequently, the 1D helices in **1-1D** transform into a 3D network with the 1D backbone along the crystallographic *a* axis maintained in the new structure. The removal of eg ligands should occur from the same side of the helices in **1-1D** to introduce the spontaneous polarization along the crystallographic *b* axis observed in **1-3D**. Similar to the first-step SCSC transformation, the strong hydrogen bonds guide the reaction and maintain the integrity of the crystal lattices in both **1-1D** and **1-3D** (Supplementary Tables 4 and 5).

Density functional theory (DFT) calculations were performed to investigate the mechanism by which the elimination of chelating eg ligands occurs during the first-step SCSC transformation. As shown in Fig. 5, the energy of **1-1D** is higher than that of **1-0D** by 0.04 eV per $[Zn^{II}(eg)_3]SO_4$ molecule. Such a small energy change, which is smaller than the energy variations for typical solid-state phase transitions[10–12,43,44], can be compensated by the entropy gain associated with the release of eg molecules during the transition. According to the calculation, the most energy-consuming step is **1-0D** → INT1 (1.02 eV uphill), where one Zn−O(eg) coordination bond is substituted by a sulfate ion. Then, the relative energy decreases to 0.84 eV upon releasing one eg molecule (INT2). This step is entropy driven, therefore the SCSC transformation manifests as an elimination of the chelating eg ligand instead of the substitution of one Zn−O(eg) coordination bond. Moreover, the calculation results reveal that the lattice energy can be further decreased by the molecular conformation shift INT2 → INT3, where half of the Zn-motif changes from Δ to Λ conformation. This result is consistent with the chiral interconversion of complex cations during the SCSC transformation observed in the experiments.

Similar 0D to 1D chiral structure transformations were also observed in the $Ni^{II}$ and $Co^{II}$ analogs of the present compound, but the 1D-to-3D structural change was not detected in these compounds, potentially because the crystal field stabilization energies of $Ni^{II}$ and $Co^{II}$ in their octahedral complexes are larger than those in their tetrahedral complexes (Supplementary Figs. 15 and 16 and Supplementary Table 6). In contrast, the analogous $Cu^{II}$ complex undergoes a different one-step centric ($P2_1/n$) to polar ($P2_1$) SCSC transformation, where two-thirds of the eg ligands are removed simultaneously (Supplementary Fig. 17 and Supplementary Table 7). Because of the significant Jahn-Teller effect of $Cu^{II}$, the metal centers after SCSC transformation are coordinated with planar geometry by two O atoms from one eg ligand and two O atoms from two sulfate dianions

(Supplementary Fig. 18). The circular building blocks with two $Cu^{II}$ metal centers bridged by two sulfate dianions are further linked into a 1D structure through subordinate Cu−O coordination bonds ($d_{Cu-Cu} > 2.3$ Å, see Supplementary Fig. 19). Like the $Zn^{II}$ compound, the strong O−H···O hydrogen-bond interactions exist in the other metal compounds before and after SCSC transformation, further evidencing the important roles played by strong hydrogen bonds in stabilizing the crystal lattices. Notably, all the SCSC transformations detected in these compounds are accompanied by a centric-to-chiral symmetric variation in the single crystals, indicating that the elimination of chelating ligands during solid-state reactions is a possible strategy for accessing new chiral materials.

In conclusion, we demonstrated an adaptive single crystal that undergoes a two-step giant SCSC transformation with large variations in volume (50%) and weight (36%) as a consequence of the thermal elimination of chelating eg ligands. Due to the robustness of their single-crystal nature, the reversible structural transformation, whereby the **1-0D** discrete structure transforms into a **1-1D** helical structure and further into a **1-3D** network structure could be fully characterized by SC-XRD analyses. The super-adaptive crystal lattices capable of giant SCSC transformation are connected to the strong hydrogen-bond interactions that guide the reaction and fasten the crystal structure. Moreover, the elimination of chelating ligands facilitates a conformational change in the complex cation that manifests as a Δ-to-Λ (or Λ-to-Δ in an enantiomeric single crystal) chiral interconversion. The structural variation of the molecules involves a centric-chiral-polar symmetric change of the bulk single crystal. This work will inspire the future development of superior flexible materials that lie on the boundary between single crystals and gels. Furthermore, the achiral-to-chiral symmetric variation induced by the elimination of chelating ligands represents a promising strategy for the preparation of new chiral materials.

## Methods

**Synthesis of $[Zn^{II}(eg)_3]SO_4$ (1-0D).** The colorless crystals of **1-0D** were prepared by a slow diffusion of acetone solvent into an eg (eg = ethylene glycol) solution (5 mL) of $ZnSO_4 \cdot 7H_2O$ (0.5 mmol). The yield is ~80% on the basis of $ZnSO_4 \cdot 7H_2O$. Anal. $C_6H_{18}O_{10}SZn$ (347.63); calcd. C: 20.73%, H: 5.22%; found C: 20.69%, H: 5.29%.

**Synthesis of $[Zn^{II}(eg)_2(\mu\text{-}SO_4)]_n$ (1-1D).** The single crystals of **1-1D** can be obtained by heating the crystals of **1-0D** at 366 K for 2 h. Anal. $C_4H_{12}O_8SZn$ (285.57); calcd. C: 16.82%, H: 4.24%; found C: 16.91%, H: 4.25%.

**Synthesis of {[Zn$^{II}$(eg)$_2$]Zn$^{II}$($\mu_3$-SO$_4$)$_2$}$_n$ (1-3D)**. The **1-3D** was prepared by further heating the crystals of **1-1D** to 410 K, and the single crystal that suitable for SC-XRD analyses was carefully selected from cracked crystals. Anal. C$_2$H$_6$O$_6$SZn (223.5); calcd. C: 10.75%, H: 2.71%; found C: 10.72%, H: 2.73%.

**Syntheses of [Co$^{II}$(eg)$_3$]SO$_4$ and [Co$^{II}$(eg)$_2$($\mu$-SO$_4$)]$_n$**. The dark-purple color crystals were prepared by a slow diffusion of acetone solvent into an eg solution (5 mL) of CoSO$_4$ (0.5 mmol). The yield is ~70% on the basis of CoSO$_4$. Anal. C$_6$H$_{18}$O$_{10}$SCo (341.19); calcd. C: 21.12%, H:5.32%; found C: 21.09%, H: 5.26%. The 1D single crystals of [Co$^{II}$(eg)$_2$($\mu$-SO$_4$)]$_n$ were obtained by heating the crystals of [Co$^{II}$(eg)$_3$]SO$_4$ to 405 K. Anal. C$_4$H$_{12}$O$_8$SCo (279.13); calcd. C: 17.21%, H: 4.33%; found C: 17.27%, H: 4.38%.

**Synthesis of [Cu$^{II}$(eg)$_3$]SO$_4$ and {[Cu$^{II}$(eg)]$_2$($\mu$-SO$_4$)$_2$}$_n$**. The blue crystals of [Cu$^{II}$(eg)$_3$]SO$_4$ were prepared with the same method as described above. The yield is ~70% on the basis of CuSO$_4$. Anal. C$_6$H$_{18}$O$_{10}$SCu (345.81); calcd. C: 20.84%, H: 5.25%; found C: 20.91%, H: 5.27%. In contrast to the compounds of Zn$^{II}$ and Co$^{II}$, two-thirds of eg ligands were simultaneously removed upon heating the [Cu$^{II}$(eg)$_3$] SO$_4$ to 397 K, producing a new 1D structure of {[Cu$^{II}$(eg)]$_2$($\mu$-SO$_4$)$_2$}. Anal. C$_2$H$_6$O$_6$SCu (221.67); calcd. C: 10.83%, H: 2.73%; found C: 10.91% H: 2.72%.

**SHG measurement**. The temperature-dependent SHG measurement was performed on polycrystals of **1-0D** with a Nd:YAG laser ($\lambda = 1064$ nm) as the fundamental light source. The generated SHG light ($\lambda(2\omega) = 532$ nm) was detected as the temperature varies.

**CD measurements**. CD measurements of individual single crystal were performed with powder sample prepared from individual large single crystal of **1-1D** (Supplementary Fig. 5) under a constant flow of nitrogen on JASCO J-810 (Fig. 3b) or JASCO J-1500 spectropolarimeter (Supplementary Fig. 11). CD measurements of racemic mixture were performed with the powder samples prepared from a large number of single crystals.

**TG measurement**. TG measurements were recorded with TG-DTA 6200 thermal analyzer (HITACHI) in a dynamic nitrogen atmosphere. The data of compound **1-0D** was controlled at a heating rate of 3, 10, and 20 K min$^{-1}$, and that of compound [Co$^{II}$(eg)$_3$]SO$_4$ and [Cu$^{II}$(eg)$_3$]SO$_4$ was collected with a heating rate of 5 K min$^{-1}$.

**Elemental analyses**. Elemental analyses of C and H were performed on an EUROVECTER EA3000 analyzer in the Analysis & Testing Center of Beijing Institute of Technology.

## Data availability

All data generated and analyzed in this study are included in the Article and its Supplementary Information, and are also available from authors upon request. The X-ray crystallographic coordinates for structures reported in this study have been deposited at the Cambridge Crystallographic Data Centre (CCDC) under deposition numbers CCDC: 2084540 for **1-0D**, 2084544 for **1-1D** with right-handedness, 2084545 for **1-1D** with left-handedness, 2084546 for **1-3D**, 2084547 for [Co$^{II}$(eg)$_3$]SO$_4$, 2084548 for [Co$^{II}$(eg)$_2$($\mu$-SO$_4$)]$_n$, 2084549 for [Cu$^{II}$(eg)$_3$]SO$_4$, and 2084550 for {[Cu$^{II}$(eg)]$_2$($\mu$-SO$_4$)$_2$}$_n$. These data can be obtained free of charge via http://www.ccdc.cam.ac.uk/data_request/cif.

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

## Acknowledgements

This work was supported by the National Natural Science Foundation of China (Grants 22071009 and 21701013 (Z.-S.Y.), 21671161 and 21971016 (J.T.), and 21903004 (J.X.)). The technical support from the staff at the Analysis & Testing Center, Beijing Institute of Technology is also appreciated.

## Author contributions

Z.-S.Y. and Y.L. designed the study and wrote the manuscript. Y.L. synthesized the materials and performed the experimental measurements. B.Z. and J.X. performed the DFT calculations. J.-P.X. assisted in the PXRD measurements. Z.-S.Y. and J.T. supervised the research. All authors discussed the results and commented on the manuscript.

## Competing interests
The authors declare no competing interests.
