## [Peer Review File · Nature Communications]

Giant single-crystal-to-single-crystal transformations associated with chiral interconversion induced by elimination of chelating ligandsREVIEWER COMMENTS

Reviewer #1 (Remarks to the Author):

In this manuscript, the authors reported a robust Zn complex exhibiting two-step reversible single-crystal-to-single-crystal (SCSC) transformation induced by successive elimination of ethylene glycol chelating ligands upon heating. The originality of this work, as claimed by the authors, lies in the first example of the SCSC transformation stimulated by the elimination of chelating ligands. The elimination of ethylene glycol ligands led to a massive decrease in the volume and mass of the crystals, accompanying the structural transformations from zero-dimensional (1-0D) to one-dimensional (1-1D) and further to three-dimensional structure (1-3D). The SCSC involves a chiral interconversion in the Zn complex, as demonstrated by single-crystal X-ray diffraction analyses and additional experiments. Although the presented work is certainly interesting from a scientific perspective, I could not see the important breakthrough expected for a publication in a high-impact journal. Therefore, I think this work does not meet the criteria required for publication in Nature Communications, and I suggest a more specialized journal.

There are minor issues that should be discussed to improve the quality of the manuscript.

1) The PXRD data (Fig. 2b) showed that the SCSC transformation from 1-0D to 1-1D almost completed at 343 K, at which the loss of ethylene glycol is still negligible from the thermogravimetric analysis (Fig. 2a). I think that authors need to give a proper reason in the main body of the manuscript to help readers understand.

2) Is there any possibility that the reversible SCSC transformations between 1-1D and 1-3D structures take place in a different way? In the transformation from 1-1D to 1-3D, two structures seem to be present simultaneously with different ratios during the transformation process (Fig. 2b). However, in the reverse SCSC transformation from 1-3D to 1-1D (Supp. Fig. 9), I am reluctant to admit the above statement. Have the authors ever considered the existence of an additional intermediate in the transformation process?

3) The authors presented the solid-state circular dichroism (CD) spectra of the single crystal of 1-1D with enantiomeric structures. Can the enantiopure 1-1D crystals be obtained experimentally from 1-0D crystals? Or, which enantiomer is present as the major conformation in the bulk state of 1-1D?

Reviewer #2 (Remarks to the Author):

The authors describe a remarkable series of single-crystal to single-crystal transformations in simple transition metal complexes of ethylene glycol with sulfate anions. The manuscript is very well written and easy to follow. As the authors have noted, this is by far the most spectacular series of single-crystal transformations involving the elimination (and reinsertion) of ligands, and the only example so far of such processes involving chelating ligands. For the most part, the results are very well supported by the appropriate experimental methods (SCD, PXRD, TGA, SHG measurements, and solid-state CD). The authors have also applied computational methods to explain the transformations. I highly recommend acceptance of this manuscript after relatively minor revision according to the following suggestions (I am using line numbers from the original manuscript - i.e., before the CIF files were regenerated):

1. where reports of single-crystal to single-crystal transformations are concerned, it is vital to eliminate the possibility of crystal dissolution and regrowth. The best-case scenario is to carry out all of the single-crystal diffraction experiments using the same crystal. Otherwise, it is useful to record time-lapse photographs of the crystals before, during and after the transformations. In a couple of the cases, the authors have indeed obtained different structures from the same crystal. However, they should provide convincing evidence that none of the transformations could possibly have involved dissolution/regrowth.

2. Line 47: replace "of porous" with "by porous"

3. Lines 47-50: the authors state that they were able to envision beforehand that elimination of

chelating ligands would lead to dramatic SCSC transformations. I could be wrong, but I am not convinced that this is entirely true and I am sure that some level of serendipity was involved. If the authors were able to predict the outcomes of their experiments, they would have truly unique capabilities. There is nothing wrong with admitting to serendipity, and it would not detract from the work at all.

4. Line 78: replace "monoclinic central-symmetry" with "centrosymmetric monoclinic"

5. Line 82: the total weight loss at 410 K is 36%, so the second weight loss event is not a "further 36%", but rather a further 18%.

6. Line 99: The space group P212121 should not be referred to as a "chiral space group". It is, of course, the space group of a chiral structure, but P212121 is its own enantiomorph (this also applies to P21). See Page 7 of "Concepts and nomenclature in chemical crystallography", Volume 5 of Supramolecular Chemistry: From Molecules to Nanomaterials, Wiley, 2011 (DOI: 10.1002/9780470661345.smc108). There are a total of 65 space groups that contain only symmetry operations that are compatible with chiral crystal structures, and these are termed the Sohncke space groups. My recommendation is to rewrite the sentence as follows: "Because the helices in each individual single crystal have the same handedness (Supplementary Fig. 6), the single crystal adopts the orthorhombic Sohncke space group P212121."

7. P21 is both a polar and a Sohncke space group. I suggest rewording the sentence starting on line 119 as follows: "Because all the eg ligands in the 3D structure orient principally in a same direction along the crystallographic b axis, the symmetry of the single crystal changes to that of the polar Sohncke space group P21, with a spontaneous polarization emerging along the crystallographic b axis." It is not necessary to mention that P21 is a monoclinic space group - the reader should know this.

8. Line 168: change "in the P21/c space group" to "in the space group P21/c".

9. Line 193: change "central-symmetry" to "centrosymmetry".

10. Lines 67, 232, 244, 259: change "central" to "centric". Also in the abstract, Part (a) of Figure 1, and the caption of Figure 3.

11. The caption of Figure 2 uses "SCSC" twice. However, these experiments were carried out on polycrystalline materials, and so cannot be said to be monitoring SCSC transformations. Please omit "SCSC" in both cases. The same goes for the caption of Figure 3 since the CD spectra were recorded for powdered material.

12. In numerous places the authors refer to "a-axis", "b-axis" and "c-axis". Using a hyphen would be correct if these were used as compound nouns (e.g., "a-axis length" where "a-axis" is the modifier of "length"), but in these particular cases the parameters a, b and c are simply the modifiers of "axis", and so the hyphens should be omitted.

13. One of the structures becomes twinned - there is no mention of this in the main text - is this significant in terms of the mechanism of the transformation?

14. Table 5 of the supplementary information states that it lists "exceptionally strong" hydrogen bonds. Given their H...O lengths (with the exception of the last entry only), the hydrogen bonds listed do not stand out as being unusually strong.

Overall, this manuscript describes an outstanding body of work and I believe that the findings would appeal to a wide audience.

Len Barbour

Reviewer #3 (Remarks to the Author):

The authors report a new Zn-based material that undergoes a transformation through two stages via thermal removal of ethylene glycol, triggering large volume changes.

The magnitude of the volume changes and the reversibility of the transitions will be key points of interest to the materials community. The retention of single crystallinity through these transitions is quite remarkable.

The chiral interconversion that accompanies the transitions is interesting although I personally can't comment on the uniqueness or significance of this in the context of other research.

The article is well constructed and clearly written, with the discussion conducted to a high standard.

The experimental methodology is appropriate and thorough and includes detailed crystallographic structural analysis as well as thermophysical and optical analysis. The mechanisms for structural changes that are presented as a result of these are plausible.

As a side note: there is some similarity here to certain clay structures which can be expanded by adding ethylene glycol.

Some suggested changes:

1. I did not see any experimental details for the thermogravimetric analysis. These should be added. The heating rate for the TG curve in Figure 2a is relevant and should be given in the caption.
2. Information about the method or source of the elemental analysis values for the compounds should also be provided. If done by an external lab, give the name of the lab.
3. For the SHG plot in Figure 3a, even if arbitrary units are used, there should still be a scale on the y-axis to show where the zero intensity value is. Common practice appears to be to normalise the intensity and use a scale of 0 to 1.
4. At line 82, "further ~36%" should say something like "further 18% for a total of 36%". The words in parentheses on line 84 seem redundant.
5. On line 211, the abbreviation "DFT" should be included after the first mention of density functional theory.

Response to reviewer:

We are grateful to the reviewers for their thorough evaluation of our manuscript "**Giant single-crystal-to-single-crystal transformations associated with chiral interconversion induced by elimination of chelating ligands**". In the revised version of our manuscript, we addressed all of the points raised by the reviewers, and we think the resulting study was much improved. Specific revisions and responses to each comment are provided in detail below.

Response to reviewer #1:

In this manuscript, the authors reported a robust Zn complex exhibiting two-step reversible single-crystal-to-single-crystal (SCSC) transformation induced by successive elimination of ethylene glycol chelating ligands upon heating. The originality of this work, as claimed by the authors, lies in the first example of the SCSC transformation stimulated by the elimination of chelating ligands. The elimination of ethylene glycol ligands led to a massive decrease in the volume and mass of the crystals, accompanying the structural transformations from zero-dimensional (1-0D) to one-dimensional (1-1D) and further to three-dimensional structure (1-3D). The SCSC involves a chiral interconversion in the Zn complex, as demonstrated by single-crystal X-ray diffraction analyses and additional experiments. Although the presented work is certainly interesting from a scientific perspective, I could not see the important breakthrough expected for a publication in a high-impact journal. Therefore, I think this work does not meet the criteria required for publication in Nature Communications, and I suggest a more specialized journal.

There are minor issues that should be discussed to improve the quality of the manuscript.

Response: Thank you for reading our manuscript and providing your valuable comments, which will help us improve it to a better scientific level. As you known, many single crystal materials manifesting SCSC transformations have been reported associated with interesting switchable magnetic, optical, and actuating properties. However, the related phenomena actuated by elimination of chelating ligands remains unexploited, although some chelating ligands are known to possess moderated coordination ability. In this study, we realized it for

the first time in this strong hydrogen-bonded crystal.

In this study, the elimination of chelating eg ligands leads to a massive volumetric variation that among the largest value that have been observed during the SCSC transformation. Similar guest solvent induced giant structural transformation were reported by Zhang (*Nature* **2018**, 557, 86-91) and Piotrowska (*Nat. Mater.* **2021**, 20, 403-409), and as reviewer 3 suggested, this result is akin to the large lattice expansion of certain clay structures induced by the adsorption of eg solvent. However, this compound is the only example that can provide straightly detailed structural insights into the giant lattice transformation. Such atomic-level structural information will be very useful to future material design.

Based on the single crystal structure, we found the elimination of chelating eg ligands induced a centric-chiral symmetric variation of the single crystal, not only in the compounds of Zn^{II} and Co^{II}, but also in the compound of Cu^{II} that undergoes a different SCSC transformation. From a structural viewpoint, this symmetry variation should closely associate with the synchronous shift of *cis*-coordination sites that facilitates the conformational conversion of molecules. Hence, the SCSC transformation actuated by elimination of chelating ligands provides a potential pathway to synthesize novel chiral compounds.

We believe that these results are sufficiently novel and intriguing to warrant publication in *Nat. Commun.*

1. The PXRD data (Fig. 2b) showed that the SCSC transformation from 1-0D to 1-1D almost completed at 343 K, at which the loss of ethylene glycol is still negligible from the thermogravimetric analysis (Fig. 2a). I think that authors need to give a proper reason in the main body of the manuscript to help readers understand.

Response: Thank you for your careful review. The different transition temperatures in PXRD and thermogravimetric measurements can be ascribed to the different testing methods. The thermogravimetric measurement was performed in a dynamic nitrogen atmosphere with a constant heating rate of 3 K min⁻¹, while the PXRD data was collected in a vacuum condition

with a settle mode that the temperature was stabilized at each set point for 30 min before test. As the structural transition of this compound is highly dependent on the heating process, the different transition temperatures were detected in PXRD and thermogravimetric measurements. This point was clarified in the main body of the manuscript and detailed experimental information was added in the Method section as below:

In the main body of the manuscript:

The transition temperature in the PXRD measurement is slightly lower than that in the TG analyses, which can be ascribed to the different heating method and test condition in the PXRD and TG measurements (for details see Method section).

In the Method section:

TG measurement: TG measurements were recorded with TG-DTA 6200 thermal analyzer (HITACHI) in a dynamic nitrogen atmosphere. The data of compound **1-0D** was controlled at a heating rate of 3, 10 and 20 K min⁻¹, and that of compound [Co^{II}(eg)₃]SO₄ and [Cu^{II}(eg)₃]SO₄ was collected with a heating rate of 5 K min⁻¹.

Powder X-ray diffraction (PXRD): PXRD was recorded on a PANalytical diffractometer with Cu K α radiation equipped with a TTK450 accessory in a temperature range from 273 K to 473 K at an interval of 25 K in a vacuum condition. The sample was stabilized at each temperature point for 30 min before measurement.

2. Is there any possibility that the reversible SCSC transformations between 1-1D and 1-3D structures take place in a different way? In the transformation from 1-1D to 1-3D, two structures seem to be present simultaneously with different ratios during the transformation process (Fig. 2b). However, in the reverse SCSC transformation from 1-3D to 1-1D (Supp. Fig. 9), I am reluctant to admit the above statement. Have the authors ever considered the existence of an additional intermediate in the transformation process?

Response: Thank you for your meticulous examination. We further measured the reverse SCSC transformation from **1-3D** to **1-1D**, and the new results were amended in the Supplementary Figure 10.

As shown in the new Figure, the structures of **1-3D** and **1-1D** present simultaneously in the transition process from **1-3D** to **1-1D**, and the peaks that denote to **1-1D** rapidly increase as the time-lapse from 15 min to 30 min, and then gradually increases with time further extended. According to this result, we think the SCSC transformations from **1-3D** to **1-1D** should take place in the same way as the structural transformation from **1-1D** to **1-3D**. Notably, we found the time consumption for the structural transition from **1-3D** to **1-1D**, and further to **1-0D** depends on the size and quality of the sample. This point was clarified in the figure caption.

Supplementary Figure 10. The reversibility of **1-3D** to **1-1D** and further to **1-0D** monitored by the in-situ PXRD. The peaks that denote to the structure of **1-1D** and **1-0D** appear after 180 and 720 min in the acetone solution of eg, respectively, verifying the reversibility of two-step structural transformation. Notably, the time consumption for the structural transition from **1-3D** to **1-1D**, and further to **1-0D** depends on the size and quality of the sample.

3. The authors presented the solid-state circular dichroism (CD) spectra of the single crystal

of 1-1D with enantiomeric structures. Can the enantiopure 1-1D crystals be obtained experimentally from 1-0D crystals? Or, which enantiomer is present as the major conformation in the bulk state of 1-1D?

Response: Thanks for your valuable comments. We performed more CD measurements to investigate the chirality of the **1-1D** crystals. The new results are shown in the Figure below. The individual single crystals show distinct CD signal. As the number of samples increases, the number of crystals with Λ molecular conformation tends to equate to that of Δ molecular conformation. Moreover, we measured the CD spectra of several powder samples prepared from a large number of single crystals. This experiment was repeated on four samples from different mixture. However, no substantial CD signal was observed. These results suggest the bulk state of **1-1D** are not enantiopure crystals, but racemic mixture that has equal amounts of left- and right-handed enantiomers. The new result was inserted in SI as Supplementary Figure 11.

Supplementary Figure 11. The CD spectra of individual single crystals and powder crystals. The number of individual single crystals with Λ molecular conformation (blue lines) tends to equate to that of Δ molecular conformation (red lines). No distinct CD signal was detected in the powder crystals prepared from a lot of single crystals. These results suggest the bulk state of **1-1D** are racemic mixture that has equal amounts of left- and right-handed

enantiomers.

Response to reviewer #2:

Comments to Authors:

The authors describe a remarkable series of single-crystal to single-crystal transformations in simple transition metal complexes of ethylene glycol with sulfate anions. The manuscript is very well written and easy to follow. As the authors have noted, this is by far the most spectacular series of single-crystal transformations involving the elimination (and reinsertion) of ligands, and the only example so far of such processes involving chelating ligands. For the most part, the results are very well supported by the appropriate experimental methods (SCD, PXRD, TGA, SHG measurements, and solid-state CD). The authors have also applied computational methods to explain the transformations. I highly recommend acceptance of this manuscript after relatively minor revision according to the following suggestions (I am using line numbers from the original manuscript - i.e., before the CIF files were regenerated):

Response: We thank the reviewer for positive comments on our work and valuable suggestions concerning the manuscript. Our point-by-point responses are provided below.

1. Where reports of single-crystal to single-crystal transformations are concerned, it is vital to eliminate the possibility of crystal dissolution and regrowth. The best-case scenario is to carry out all of the single-crystal diffraction experiments using the same crystal. Otherwise, it is useful to record time-lapse photographs of the crystals before, during and after the transformations. In a couple of the cases, the authors have indeed obtained different structures from the same crystal. However, they should provide convincing evidence that none of the transformations could possibly have involved dissolution/regrowth.

Response: Thank you for your suggestion. We tried several times to carry out all of the single-crystal diffraction experiments using the same crystal, however, only **1-0D** to **1-1D** was succeed. For the structural transition from **1-1D** to **1-3D**, only the cell parameters of structure **1-3D** can be reduced from the diffraction spots. Therefore, the single crystal of **1-3D** that suitable for SC-XRD analyses was carefully selected from cracked crystals. According to your suggestion, we recorded the time-lapse photographs of the crystals before, during and after the transformations,

and these photographs were inserted in the SI as Supplementary Figure 4.

Supplementary Figure 4. The time-lapse photographs of the crystals before, during and after the transformations from **1-0D** to **1-1D**, and further to **1-3D**. The crystalline shape of the sample retained during the phase transition, eliminating the possibility of crystal dissolution and regrowth during the structural transformation.

2. Line 47: replace "of porous" with "by porous".

Response: The "of porous" was replaced with "by porous".

3. Lines 47-50: The authors state that they were able to envision beforehand that elimination of chelating ligands would lead to dramatic SCSC transformations. I could be wrong, but I am not convinced that this is entirely true and I am sure that some level of serendipity was involved. If the authors were able to predict the outcomes of their experiments, they would have truly unique capabilities. There is nothing wrong with admitting to serendipity, and it would not detract from the work at all.

Response: Yes, you are right. The prediction of the experimental outcomes of a molecular system that highly relies on the flexible intermolecular interactions remains a challenge. In the

present example, the large structural change induced by the elimination of chelating ligands can be expected, but the resulting single-crystal-to-single-crystal transformation is beyond expectation. Hence, the expression “we envisioned” is inaccurate. This sentence was changed as below:

In addition to the guest sorption by porous structures, the elimination of chelating ligands from a nonporous crystal might also lead to a dramatic SCSC transformation because two coordination sites are released concertedly during the reaction.

And the word “Serendipitously” was added in the result summary part of the introduction section to clarify this point:

Serendipitously, although the structural change is exceptionally large, the single-crystal nature persists because of the strong hydrogen-bond interactions between the complex cations and sulfates.

4. Line 78: replace "monoclinic central-symmetry" with "centrosymmetric monoclinic".

Response: The "monoclinic central-symmetry" was replaced with "centrosymmetric monoclinic".

5. Line 82: the total weight loss at 410 K is 36%, so the second weight loss event is not a "further 36%", but rather a further 18%.

Response: Thank you for your careful examination. We have corrected this sentence according to the suggestion from you and reviewer 3 as below:

The weight loss of ~18% in the first step (344 to 378 K) and further ~18% (390 to 410 K) in the second step (for a total of 36%) indicate that two of the three eg ligands are removed one-by-one in the two-step weight loss process (the corresponding calculated weight-loss values are 18% in each step).

6. Line 99: The space group $P2_12_12_1$ should not be referred to as a "chiral space group". It is, of course, the space group of a chiral structure, but $P2_12_12_1$ is its own enantiomorph (this

also applies to $P2_1$). See Page 7 of "Concepts and nomenclature in chemical crystallography", Volume 5 of Supramolecular Chemistry: From Molecules to Nanomaterials, Wiley, 2011 (DOI: 10.1002/9780470661345.smc108). There are a total of 65 space groups that contain only symmetry operations that are compatible with chiral crystal structures, and these are termed the Sohncke space groups. My recommendation is to rewrite the sentence as follows: "Because the helices in each individual single crystal have the same handedness (Supplementary Fig. 6), the single crystal adopts the orthorhombic Sohncke space group $P2_12_12_1$."

Response: Thank you for your recommendation. We have changed the sentence according to your suggestion, and the literature you recommend has been cited in the revised manuscript as Ref. 37:

37 Barbour, L. J., Das, D., Jacobs, T., Lloyd, G. O. & Smith, V. J. in Supramolecular Chemistry: From Molecules to Nanomaterials (ed Philip Gale; Jonathan Steed) (John Wiley & Sons, Ltd., 2012) 2869-2904.

7. $P2_1$ is both a polar and a Sohncke space group. I suggest rewording the sentence starting on line 119 as follows: "Because all the eg ligands in the 3D structure orient principally in a same direction along the crystallographic b axis, the symmetry of the single crystal changes to that of the polar Sohncke space group $P2_1$, with a spontaneous polarization emerging along the crystallographic b axis." It is not necessary to mention that $P2_1$ is a monoclinic space group - the reader should know this.

Response: Thanks again for your suggestion. We have changed the sentence according to your suggestion.

8. Line 168: change "in the $P2_1/c$ space group" to "in the space group $P2_1/c$ ".

Response: The sentence has been modified.

9. Line 193: change "central-symmetry" to "centrosymmetry".

Response: Modified according to your suggestion.

10. Lines 67, 232, 244, 259: change "central" to "centric". Also in the abstract, Part (a) of Figure 1, and the caption of Figure 3.

Response: The word “central” was corrected to “centric” in both the manuscript, SI and Figure 1.

11. The caption of Figure 2 uses "SCSC" twice. However, these experiments were carried out on polycrystalline materials, and so cannot be said to be monitoring SCSC transformations. Please omit "SCSC" in both cases. The same goes for the caption of Figure 3 since the CD spectra were recorded for powdered material.

Response: Thanks for your examination, we have modified the manuscript and SI according to your suggestion.

12. In numerous places the authors refer to "a-axis", "b-axis" and "c-axis". Using a hyphen would be correct if these were used as compound nouns (e.g., "a-axis length" where "a-axis" is the modifier of "length"), but in these particular cases the parameters a, b and c are simply the modifiers of "axis", and so the hyphens should be omitted.

Response: Thanks for your suggestion. the related hyphens in the manuscript and SI were deleted.

13. One of the structures becomes twinned - there is no mention of this in the main text - is this significant in terms of the mechanism of the transformation?

Response: The structural twinning was found in the crystals of $[\text{Co}^{\text{II}}(\text{eg})_2(\mu\text{-SO}_4)]_n$ and $\{[\text{Cu}^{\text{II}}(\text{eg})_2(\mu\text{-SO}_4)_2]_n\}$, and this phenomenon is usually associated with multi-domains produced during the symmetry-breaking related solid-state structural transitions, as a consequence of competition between enantiomeric structures (See *J. Am. Chem. Soc.* **134**, 11044-11049 (2012); *Angew. Chem. Int. Ed.* **56**, 15950-15953 (2017).) Accordingly, the twinning of $[\text{Co}^{\text{II}}(\text{eg})_2(\mu\text{-SO}_4)]_n$ and $\{[\text{Cu}^{\text{II}}(\text{eg})_2(\mu\text{-SO}_4)_2]_n\}$ can be ascribed to the competition between the left- and right-handedness of enantiomeric structures, and the mechanism of the transformation in each domain is not affected. This information has been added in the SI as

below:

The structures of $[\text{Co}^{\text{II}}(\text{eg})_2(\mu\text{-SO}_4)]_n$ and $\{[\text{Cu}^{\text{II}}(\text{eg})]_2(\mu\text{-SO}_4)_2\}_n$ were refined with TWIN and BASF instructions because the twin domains formed during the symmetry-breaking structural transformations. (Supplementary Methods)

Notably, the crystal twinning was detected in the structures $[\text{Co}^{\text{II}}(\text{eg})_2(\mu\text{-SO}_4)]_n$ and $\{[\text{Cu}^{\text{II}}(\text{eg})]_2(\mu\text{-SO}_4)_2\}_n$ because of the competition between the left- and right-handedness of enantiomeric structures during symmetry-breaking structural transformation^{20,21}.

(Supplementary Table 7)

References:

20 Zhang, Y. et al. Ferroelectricity Induced by Ordering of Twisting Motion in a Molecular Rotor. *J. Am. Chem. Soc.* **134**, 11044-11049 (2012).

21 Phillips, A. E. & Fortes, A. D. Crossover between Tilt Families and Zero Area Thermal Expansion in Hybrid Prussian Blue Analogues. *Angew. Chem. Int. Ed.* **56**, 15950-15953 (2017).

14. Table 5 of the supplementary information states that it lists "exceptionally strong" hydrogen bonds. Given their H...O lengths (with the exception of the last entry only), the hydrogen bonds listed do not stand out as being unusually strong.

Response: You are right, the hydrogen bond interactions in these compounds are strong, but not "exceptionally strong", we have modified this inaccurate expression in the manuscript and SI.

Overall, this manuscript describes an outstanding body of work and I believe that the findings would appeal to a wide audience.

Response: Thanks again for your positive comments and careful examination to improve the quality of our manuscript.

Response to reviewer #3:

The authors report a new Zn-based material that undergoes a transformation through two stages via thermal removal of ethylene glycol, triggering large volume changes.

The magnitude of the volume changes and the reversibility of the transitions will be key points of interest to the materials community. The retention of single crystallinity through these transitions is quite remarkable.

The chiral interconversion that accompanies the transitions is interesting although I personally can't comment on the uniqueness or significance of this in the context of other research.

The article is well constructed and clearly written, with the discussion conducted to a high standard.

The experimental methodology is appropriate and thorough and includes detailed crystallographic structural analysis as well as thermophysical and optical analysis. The mechanisms for structural changes that are presented as a result of these are plausible.

As a side note: there is some similarity here to certain clay structures which can be expanded by adding ethylene glycol.

Response: Thank you for reviewing our manuscript and providing valuable comments to improve it. Indeed, the giant structural transformation in this compound is some similarity to certain clay structures which can be expanded by adding ethylene glycol. However, the SCSC transformation in present compound can provide detailed structural insights into the giant structural change, which may also help understanding the giant ductility of the clay structures.

This point was mentioned in the manuscript as below:

The 50% total volumetric shrinkage and 36% total weight loss observed for the present material, which are reminiscent of the expansion of clay structures by adding ethylene glycol^{39,40}, are larger than those observed for other typical single-crystal materials that undergo reversible SCSC transformations,

39 Mosser-Ruck, R., Devineau, K., Charpentier, D. & Cathelineau, M. Effects of ethylene glycol saturation protocols on XRD patterns: a critical review and discussion. *Clay. Clay. Miner.* **53**, 631-638 (2005).

40 Kakuta, T., Baba, Y., Yamagishi, T.-a. & Ogoshi, T. Supramolecular exfoliation of layer silicate clay by novel cationic pillar[5]arene intercalants. *Sci. Rep.* **11**, 10637 (2021).

1. Comments: I did not see any experimental details for the thermogravimetric analysis. These should be added. The heating rate for the TG curve in Figure 2a is relevant and should be given in the caption.

Response: Thank you for your suggestion. We have added the experimental details for thermogravimetric analysis in the Methods section of manuscript as below:

TG measurement: TG measurements were recorded with TG-DTA 6200 thermal analyzer (HITACHI) in a dynamic nitrogen atmosphere. The data of compound **1-0D** was controlled at a heating rate of 3, 10 and 20 K min⁻¹, and that of compound [Co^{II}(eg)₃]SO₄ and [Cu^{II}(eg)₃]SO₄ was collected with a heating rate of 5 K min⁻¹.

And the heating rate was added in the caption of Figure 2a:

(a) TG curve of compound **1-0D** with a heating rate of 3 K min⁻¹.

2. Comments: Information about the method or source of the elemental analysis values for the compounds should also be provided. If done by an external lab, give the name of the lab.

Response: Thanks again for your suggestion. The experimental was added in the Methods section as below:

Elemental analyses: Elemental analyses of C and H were performed on an EUROVECTER EA3000 analyzer in the Analysis & Testing Center of Beijing Institute of Technology.

3. For the SHG plot in Figure 3a, even if arbitrary units are used, there should still be a scale on the y-axis to show where the zero intensity value is. Common practice appears to be to normalise the intensity and use a scale of 0 to 1.

Response: Thank you for your suggestion. the scale of normalized intensity was added in Figure 3a as below:

Figure 3.

4. At line 82, “further ~36%” should say something like “further 18% for a total of 36%”. The words in parentheses on line 84 seem redundant.

Response: According to the comments from you and reviewer 2, the wrong expression has been modified as below:

The weight loss of ~18% in the first step (344 to 378 K) and further ~18% (390 to 410 K) in the second step (for a total of 36%) indicate that two of the three eg ligands are removed one-by-one in the two-step weight loss process (the corresponding calculated weight-loss values are 18% in each step).

5. Comments: On line 211, the abbreviation “DFT” should be included after the first mention of density functional theory.

Response: Added according to your suggestion.

REVIEWERS' COMMENTS:

Reviewer #1 (Remarks to the Author):

All of my minor concerns have been well answered, but I'm still a bit hesitant to recommend publication of this work.

Reviewer #2 (Remarks to the Author):

I have carefully checked the responses to suggestions by all three reviewers. I believe that the authors have provided appropriate responses in all cases and that the manuscript has been improved considerably. I recommend that it be accepted in its current form.

Reviewer #3 (Remarks to the Author):

I appreciate the authors taking on board the comments from all of the reviewers. The manuscript is significantly improved and I see no barrier to publication.

Reviewer #1:

All of my minor concerns have been well answered, but I'm still a bit hesitant to recommend publication of this work.

Reviewer #2:

I have carefully checked the responses to suggestions by all three reviewers. I believe that the authors have provided appropriate responses in all cases and that the manuscript has been improved considerably. I recommend that it be accepted in its current form.

Reviewer #3:

I appreciate the authors taking on board the comments from all of the reviewers. The manuscript is significantly improved and I see no barrier to publication.

Response: We would like to express our gratitude once again to all reviewers for providing insightful comments concerning our manuscript.